# Exercise-Induced Proteomic Profile Changes in Patients with Advanced Heart Failure

**DOI:** 10.3390/biomedicines12102267

**Published:** 2024-10-05

**Authors:** Anna Drohomirecka, Joanna Waś, Ewa Sitkiewicz, Bianka Świderska, Anna Lutyńska, Tomasz Rywik, Tomasz Zieliński

**Affiliations:** 1Department of Heart Failure and Transplantation, National Institute of Cardiology, Alpejska 42, 04-628 Warsaw, Polandtzielinski@ikard.pl (T.Z.); 2Department of Medical Biology, National Institute of Cardiology, Alpejska 42, 04-628 Warsaw, Poland; 3Mass Spectrometry Laboratory, Institute of Biochemistry and Biophysics, Polish Academy of Sciences, 5a Pawinski Street, 02-106 Warsaw, Poland; ewa@ibb.waw.pl (E.S.);

**Keywords:** heart failure, exercise intolerance, proteomic profile, exerkines

## Abstract

Background/Objectives: The pathophysiological background of the processes activated by physical activity in patients with heart failure (HF) is not fully understood. Proteomic studies can help to preliminarily identify new protein markers for unknown or poorly defined physiological processes. We aimed to analyse the changes in the plasma proteomic profile of HF patients after a cardiopulmonary exercise test (CPET) to define pathways involved in the response to exercise. Methods: The study prospectively enrolled 20 male patients with advanced HF (aged 53.3 ± 8.3 years). Blood samples were taken from the patients before and immediately after the CPET to obtain plasma proteomic profiles. Two-sample *t*-tests (paired or non-paired) were performed with and without false discovery rate (FDR) correction for multiple testing. Enrichment analysis was performed to associate biological processes and pathways with the study results. Results: A total of 968 plasma proteins were identified, of which 722 underwent further statistical analysis. Of these, 236 proteins showed differential expression when comparing all plasma samples collected before and after CPT (*p* < 0.05), and for 86 of these the difference remained statistically significant after FDR correction. Proteins whose expression changed after exercise are mostly involved in immune response and inflammatory processes, coagulation, cell adhesion, regulation of cellular response to stimulus and regulation of programmed cell death. There were no differences in resting proteomics according to HF etiology (ischemic vs. non-ischemic). Conclusions: Changes in the proteomic profile revealed a complexity of exercise-induced processes in patients with HF, suggesting that few major physiological pathways are involved. Further studies focusing on specific pathways are needed.

## 1. Introduction

Heart failure (HF) affects 1–2% of adults in developed countries [1]. Despite the continuous development of medical therapies, HF remains a condition with a poor prognosis, associated with significant impairment of quality of life.

HF is not a single pathological diagnosis. Nonetheless, regardless of its etiology, all HF cases are characterized by elevated intracardiac pressures and/or inadequate cardiac output at rest and/or during exercise [1]. Furthermore, as a clinical syndrome, chronic HF is accompanied by typical signs and symptoms, of which exercise intolerance with muscle fatigue and dyspnea is always a hallmark. Although hemodynamic alternations with impaired cardiac reserve followed by reduced peripheral and respiratory skeletal muscle perfusion are an underlying mechanism of impaired exercise capacity in HF, many other factors contribute. Poor pump function activates numerous neurohumoral effects that modify the physiological processes in the peripheral systems. Phillips et al. [2] described contributors to exercise limitation in HF, stressing the role of impaired metabolic vasodilation of exercising muscles. The following have been mentioned as potential mechanisms leading to enhanced vasoconstrictor tone and, consequently, to decreased tissue perfusion: increased sympathetic nervous system activation, activation of the renin–angiotensin system, impaired endothelial-dependent vasodilation, reduced blood vessel density, and increased arterial stiffness [2]. The skeletal muscles per se are also involved in the pathophysiology of HF. Several anatomical and biochemical features have been defined in the course of HF-related skeletal myopathy [3], of which mitochondrial dysfunction is responsible for the majority of the reduced energy supply [3]. Another key player in exercise intolerance in patients with HF are disturbances in respiratory system functioning due to perfusion and ventilation abnormalities [2]. Pulmonary edema (including subclinical fluid retention), increased pulmonary pressure and vascular resistance, decreased elastic recoil of the lungs, inspiratory muscle weakness, and ventilation–perfusion mismatch—all jeopardize gas exchange and oxygen utilization by the mitochondria [2,3].

The above mentioned processes show how complex the interrelationships are between the various organs and systems involved in response to physical activity. Many of them are still not yet fully understood. Numerous cells and tissues secrete signaling moiety in response to acute exercise. Exercise-induced peptides, metabolites, DNA, mRNA, microRNA and other RNA species that exhibit autocrine, paracrine and/or endocrine activity are collectively referred to as exerkines [4]. The last few years have brought new insights into exerkine research with important contributions from “omics” technology [5,6,7,8] revealing a broad spectrum of plasma proteome changes after acute exercise compared to rest. However, to the best of our knowledge, no published studies have addressed plasma proteome kinetics in patients with HF as a result of physical exertion.

Therefore, the aim of the current study was to investigate whether there are large-scale plasma proteomic changes in patients with advanced HF after acute extensive exercise and to to define the pathways involved in response to exercise.

## 2. Material and Methods

### 2.1. Study Population and Design

#### 2.1.1. Patients

This is a prospective study conducted at a tertiary cardiology center.

The study group consisted of 20 male patients with advanced chronic HF admitted to our center for an evaluation of heart transplant candidacy.

We randomly selected 10 men with ischemic and 10 men with non-ischemic etiologies of HF from a population of 51 patients examined in our earlier study [9], the inclusion criteria of which were: patients aged 18–70 years with advanced heart failure with reduced left ventricular ejection fraction (LVEF; less than 40%) who have clinical indication to undergo a cardiopulmonary exercise test (CPET) and right heart catheterization. Therefore, the exclusion criteria were as previously described [9]: administration of catecholamines, contraindications to performance of CPET, pneumonia or bronchitis within last two weeks, or severe ventilation disorders with forced expiratory volume in one second (FEV1) < 50%.

#### 2.1.2. Cardiopulmonary Exercise Test [9] (As Described Previously)

Symptom-limited submaximal CPET was performed in all patients as a part of diagnostic process and was used to estimate exercise capacity. Peak oxygen consumption was measured with a CPET on a bicycle ergometer (product of Lode Medical Technology, Groningen, The Netherlands). System calibration was performed prior to each test. The exercise test consisted of three minutes of unloaded pedaling followed by a graded increase in workload of 10 watts per minute. All tests were stopped at the request of the patients on the basis of their symptoms (severe dyspnea or fatigue). Peak VO2 was defined as the highest 30-s mean value during the period immediately before the end of exercise. Age- and sex-adjusted peak VO2 was automatically calculated by the system software (MetaLyzer 3B-R2, Cortex, Leipzig, Germany) After exercise was completed, patients spent at least 2 min in a cool-down period on the bicycle without workload. The expiratory exchange ratio [RER, defined as carbon dioxide output (VCO2) divided by oxygen consumption (VO2)] and VE/VCO2 slope (the relation between minute ventilation and carbon dioxide production) were calculated for diagnostic purposes.

#### 2.1.3. Blood Sampling and Storage

Venous blood samples were collected at rest and after completion of the CPET (after a resting period of 10 to 15 min). Plasma was frozen immediately after centrifugation and stored at −80 °C until analysis.

### 2.2. Laboratory Methods

#### 2.2.1. Protein Digestion

Internal standard (IS) samples were prepared by mixing equal volumes of plasma from each patient. Samples and IS were immunodepleted of the top 14 plasma proteins using High-Select Top 14 Abundant Protein Depletion Resin (Thermo Fisher Scientific; Waltham, MA, USA) according to the manufacturer’s protocol. Protein concentrations were measured using the Pierce™ Quantitative Colorimetric Peptide Assay (Thermo Fisher Scientific, Waltham, MA, USA).

25 µg of protein from each sample and IS was dried in a SpeedVac and reconstituted in a digestion buffer containing 5% trifluoroethanol, 5 mM tris(2-carboxyethyl)phosphine (TCEP) and 100 mM triethylammonium bicarbonate (TEAB). After 1 h incubation at 60 °C, the cysteines were blocked with 10 mM methyl methanethiosulfonate (MMTS). Proteins were digested overnight at 37 °C with 2 µg trypsin (Promega GmbH, Mannheim, Germany). Peptides were labeled with TMTpro 16plex tags (Thermo Fisher Scientific, Waltham, MA, USA) in 40 µL of acetonitrile for 1 h on vortex. Three labeling kits were used for the experiment, with 126 tags used to label the IS sample on each TMTpro set. The reaction was quenched by the addition of 8 µL of 5% hydroxylamine. After checking the labeling efficiency, the peptides were combined within each TMTpro set and desalted using 30 mg Oasis HLB columns (Waters Corporation; Milford, MA, USA). Briefly, the cartridges were preconditioned with 1 mL methanol and washed with 1 mL 1.5% acetonitrile (ACN) and 0.1% formic acid (FA). After sample loading and rinsing with 1 mL 1.5% ACN and 0.1% FA, peptides were eluted from the columns with 400 µL 90% ACN and 0.1% FA. Aliquots were dried in a SpeedVac and resuspended in 500 µL of 10 mM ammonium hydroxide with 2% ACN.

#### 2.2.2. Reversed-Phase Peptide Fractionation at High pH

TMT-labeled peptides were fractionated using high-pH reversed-phase chromatography on an XBridge Peptide BEH C18 column (4.6 × 250 mm, 130 Å, 5 µm, Waters). Separation was performed at a flow rate of 0.8 mL/min for 100 min on a Waters Acquity UPLC H-class system. The mobile phases consisted of water (A), acetonitrile (B) and 100 mM ammonium hydroxide solution (C). The percentage of phase C was maintained at 10% throughout the gradient. Fractions were collected every 1 min starting from the fourth minute of the run. The following gradient was used: 5 to 8% solvent B for 3 min, 8 to 15% for 17 min, 15 to 25% for 25 min, 25 to 33% for 15 min, 33 to 50% for 16 min, 50 to 90% for 8 min, 4 min isocratic hold at 90% and final column equilibration at 3% phase B for 2 min. The peptide elution profile was monitored at 214 nm. A total of 96 fractions were dried in Speedvac and reconstituted in Evosep solvent A (0.1% FA in water) by vortexing and sonication for 30 min each. Samples were concatenated by mixing every 48 fractions.

#### 2.2.3. Mass Spectrometry

Fractions were analyzed using an LC-MS system consisting of an Evosep One (Evosep Biosystems, Odense, Denmark) directly coupled to an Orbitrap Exploris 480 mass spectrometer (Thermo Fisher Scientific, Bremen, Germany). Concatenated peptide fractions were applied to disposable Evotips C18 trap columns (Evosep Biosystems, Odense, Denmark) as previously described [10]. Chromatography was performed at a flow rate of 500 nL/min using the 44 min (30 samples per day) preformed gradient on the EV1106 analytical column (Dr. Maisch C18 AQ, 1.9 µm beads, 150 µm ID, 15 cm long, Evosep Biosystems, Odense, Denmark). Data were acquired in positive mode using a data dependent method with the following parameters: MS1 resolution was set to 60,000 with a normalized AGC target of 300%, auto maximum inject time, and a scan range of 300 to 1700 *m*/*z*. For MS2 the resolution was set to 30,000 with a standard normalized AGC target, auto maximum inject time and the top 25 precursors within a 1.2 *m*/*z* isolation window were considered for MS/MS analysis. Dynamic exclusion was set to 20 s with a mass tolerance of ±10 ppm, a precursor intensity threshold of 5 × 10^3^ and a Precursor Fit threshold of 70%. Precursors were fragmented in HCD mode with normalized collision energy of 30%. TurboTMT resolution mode was set to “TMTpro Reagents”. Spray voltage was set to 2.1 kV, funnel RF level to 40, and heated capillary temperature to 275 °C.

## 3. Analytical Methods

### 3.1. Data Analysis

Offline recalibration, as well as peptide and protein identification, were performed in the MaxQuant/Andromeda software suite (version 2.0.1.0) [11] using the Homo sapiens Swissprot database (version 202208, canonical) and the MaxQuant contaminant database (version 2.0.1.0). The search included tryptic-generated peptides, with Metylthio (C) set as a fixed modification and Oxidation (M) as a variable modification. Reporter MS2 TMTpro 16-plex quantification was specified in order to obtain values for quantitative analysis. The reverse database was used for target/decoy statistical results validation, and peptide and protein FDR was set to 0.01. The isobaric match between runs function was enabled.

Protein groups along with quantitative data were further analyzed in Perseus (version 1.6.15) [12]. Hits from the reverse database, proteins identified only by site, and contaminants were removed. Data were normalized using the loading factor (LF) method. LF coefficients for each TMTpro channel were obtained by dividing the sum of the signal for all samples by the sum of the signal for proteins within a given channel. Intensity values for each protein within a sample were multiplied by the LF factor for that channel. Data were logarithmized, and proteins that were not identified in at least 60% of the samples were removed. The remaining missing data were imputed within the normal distribution of intensity, shifted toward the lowest values. A procedure was then performed to remove the batch effect of the TMT set on the signal intensity for individual samples (remove batch effect, Limma R algorithm). The validity of the procedure was checked using data distributions, Person correlation factors and PCA analysis.

### 3.2. Protein Identification

A total of 968 plasma proteins were identified, of which 772 were suitable for statistical analysis. The complete list of the 772 proteins that were subjected to analysis can be found in the Appendix A.

### 3.3. Statistical Analysis

Two-sample *t*-tests were used to compare differences between continuous variables (paired or non-paired, as appropriate). The tests were performed with and without false discovery rate (FDR) correction for multiple testing.

### 3.4. Gene Enrichment Analysis

Gene-encoding proteins identified as significantly altered (with FDR correction) in response to CPET were selected for enrichment analysis to link them to underlying molecular pathways and functional categories defined by gene ontology (GO). Enrichment analysis was performed with the 0.80 version of ShinyGO^®^—a web application http://bioinformatics.sdstate.edu/go/) (accessed on 1 June 2024) [13].

## 4. Results

The mean age of the patients was 53.3 ± 8.3 years (range 36 to 64 years). All patients presented with heart failure with reduced ejection fraction (HFrEF) and advanced symptoms (New York Heart Association (NYHA) functional class II–III).

Pharmacological treatment was prescribed according to the European Society of Cardiology guidelines in force at the time of enrollment.

Table 1 provides further details on patient characteristics.

A comparison of pre- and post-exercise plasma samples yielded 236 differential proteins (*p* < 0.05), of which 86 proteins met the criterion after correction for multiple testing.

Comparison of post- CPET to pre-CPET samples in patients with ischemic etiology of HF yielded 215 differential proteins (*p* < 0.05) of which 22 were found after FDR correction (q < 0.05). For the remaining comparisons (listed below), no proteins were found that met the criterion for statistical significance after adjustment for multiple testing.

When comparing post-CPET vs. pre-CPET samples in patients with non-ischemic etiology of HF, 66 differential proteins were detected. Pre-CPET samples from patients with ischemic vs. non-ischemic etiology differed in 45 proteins, and post-CPET samples differed in 14 proteins between these groups (*p* < 0.05).

For a detailed list of the identified proteins, including the differential proteins, see the Appendix A.

### Enrichment Analysis

Detailed data are shown in the Appendix A and presented as graphs, divided into three types of analyses regarding molecular function, biological process, cellular component (Figure 1, Figure 2 and Figure 3; (http://bioinformatics.sdstate.edu/go/) accessed on 1 June 2024 [13]) and interactive enrichment network graph (Figure 4, (http://bioinformatics.sdstate.edu/go/://bioinformatics.sdstate.edu/go/) accessed on 1 June 2024 [13]).

## 5. Discussion

Acute exercise results in an extensive effort of multiple organs and regulations systems, involving not only the skeletal muscle and cardiopulmonary system, but also the neural and endocrine systems. Previous studies [5,6,7] have showed a broad spectrum of changes in plasma proteome provoked by a single bout of exercise. In designing our study, we had to address some important methodological issues that could significantly affect the results. The first was to determine the optimal time to collect a blood sample after exercise. As shown by Mi et al. [6], the change in the proteome induced by exercise decreases with time, and one hour after peak exercise only 7% of the proteins still present a changed expression. At the same time, new sets of proteins with altered expression are discovered, but they are not as numerous. Because we wanted to define the response to acute exercise in HF patients, we decided to take blood samples during the resting period, no later than 15 min after completion of the CPT. In this way, we minimized the risk of unanticipated injury associated with needle punctures (this can happen when taking blood while exercising or while standing) and were able to detect the majority of proteomic changes. The second issue is natural fluctuations in the proteome. Contrepois et al. [5] compared changes observed after symptom-limited cardiopulmonary exercise with changes observed in individuals without intervention in 1-h and 24-h windows. They concluded that less than 2% of the analytes that changed with exercise also varied within an hour without exercise, but usually with a different magnitude or trajectory. In addition, they found that exercise-induced changes were significantly more pronounced (at least 2-fold greater) than individual spontaneous inter-day variation within 24 h. These findings support our assumption that the natural intra-personal variability of the proteome may be marginalized in the context of exercise-induced changes. Moreover, we included patients who were clinically stable with no symptoms of worsening HF to reduce the risk that the observed results were a consequence of disease exacerbation. Finally, our concern was the size of the study population. Proteomic studies have been performed on even smaller samples (e.g., 12 individuals [7], six men [8]) and such a group size is acceptable. We tried to select a more homogeneous population rather than a large one, especially since it has been shown that the inter-individual variability in response to an acute bout of exercise was more evident for proteins than, for example, for metabolites or transcripts [5].

In our study, 86 proteins were found to be differentially expressed after CPET (after correction for multiple testing). This number is lower than that observed in most studies in subjects without heart failure [5,6,7], although Kurgan et al. [8] identified only 20 proteins whose expression changed after acute exercise. There is no simple explanation for such a large difference between studies. In the case of our population, a smaller number of differential proteins may be related to the intensity of the exercise performed by the patients. Guseh et al. [7] found that although most proteins that changed their expression after moderate intensity exercise remained over- or under-expressed after high intensity exercise, the number of differential proteins increased significantly (from 184 to 598) with increasing exercise intensity. The individuals enrolled in the previous studies were of normal physical capacity. In contrast, we focused on patients with significant exercise intolerance due to heart failure—all study participants were classified as at least New York Heart Association II class and were only able to achieve approximately 35% of the predicted peak oxygen consumption.

Enrichment analysis of our results showed the strongest correlation of the differential proteins with regulation of coagulation/homeostasis, wound healing, protein activation cascade, body fluid levels and cell adhesion. Particularly noteworthy is the discussion of proteins involved in the regulation of coagulation processes. Changes in the expression of procoagulant and fibrinolytic proteins were observed in studies of subjects without heart failure [5,6], but in patients with heart failure (our study population) they dominated the results of the enrichment analysis. Acute exercise increased fibrinolysis in healthy individuals in a manner dependent on the intensity and duration of the exercise [14]. Furthermore, the fibrinolytic response to exercise seemed to be similar in patients with cardiovascular disease [14]. However, patients with cardiovascular disease showed an increased coagulation potential both at rest and after exercise compared to healthy subjects, involving elevated thrombin–antithrombin complexes and von Willebrand factor [14]. In our study, we observed a significant upregulation of proteins with procoagulant properties, such as von Willebrand factor (similar to the results presented previously [5,6]), fibrinogen, and coagulation factor XII; but surprisingly, at the same time, other procoagulants such as thrombospondin-1 (which has been shown to increase in individuals without heart failure [6]) or coagulation factor XI were downregulated. In addition, we also demonstrated an elevated expression of platelet glycoprotein V. This may indicate that acute, intense exercise increases prothrombotic activity in patients with heart failure, but the extent and consequences of this effect should be further defined. Moreover, Mi et al. [6] found that there was no impact on coagulation during peak exercise as opposed to 1 h after exercise. This suggests that changes in coagulation pathways occur during the recovery period, which is consistent with our results.

An interesting finding showed an interactive enrichment network. It revealed that the PI3K-Akt pathway is involved in the response to intense exercise in HF patients. One of the proteins associated with the PI3K-Akt pathway is interleukin-6. It plays a dual role in inflammatory processes—depending on how it binds to its receptor, it can have proinflammatory or anti-inflammatory effects [15]. The soluble receptor (sIL-6R), upon binding interleukin-6, is capable of inducing intracellular signaling via gp130, referred to as IL-6 trans-signaling, and does not require the presence of Il-6R on the cell membrane [15]. Trans-signaling is responsible for proinflammatory response. In addition, Zegeye et al. [15] showed that Il-6R trans-signaling activates the JAK/STAT3 pathway, along with activation of the PI3K/AKT and MEK/ERK pathways. In turn, Ouwerkerk et al. [16] recently published the results of an analysis of the BIOSTAT-CHF (Systems BIOlogy Study to TAilored Treatment in Chronic Heart Failure) study data using a specifically designed machine learning approach, where they described that 4 major pathways were associated with all-cause mortality in HF patients: (1) the PI3K/Akt pathway; (2) the MAPK pathway; (3) the Ras pathway; and (4) epidermal growth factor receptor tyrosine kinase inhibitor resistance. All of the above indicate that the decrease in interleukin-6 receptor subunit alfa after exercise in HF patients found in our study is not just a coincidental finding, but an expression of real pathophysiological processes.

It is noteworthy that there was a change in the expression of C1q and tumor necrosis factor-related protein 1 (CTRP1), which has been the subject of rather limited study thus far in the context of HF. Yang et al. [17] demonstrated that the levels of CTRP1 in the plasma and epicardial adipose tissue were elevated in patients with heart failure (HF) relative to control subjects. Higher CTRP1 levels were identified as a predictor of a worse prognosis (composite end point: cardiac death and readmission for worsening of heart failure). Furthermore, the experimental model of cell culture demonstrated that CTRP1 increases the release of aldosterone and interleukin 6. The increased expression of CTRP1 following exercise may contribute to the initiation and propagation of inflammatory processes and the activity of the renin–angiotensin system.

### 5.1. Clinical Implications

One of the primary symptoms of heart failure is exercise intolerance. The prescription of appropriate levels of physical activity, and the selection of appropriate forms of exercise within those levels, represents an inherently complex issue in the management of patients with HF. While excessive physical activity may potentially exacerbate the symptoms of HF, regular physical training has been demonstrated to enhance the quality of life and reduce cardiovascular and all-cause hospitalizations [3,18]. Both European [1] and American guidelines [19] recommend exercise training for clinically stable patients. Nevertheless, there is currently no single training protocol that is dedicated to HF patients with reduced LVEF. The parameters, such as frequency, intensity, duration, type, volume, and progression of training, should be tailored to the individual patient [18]. The proteomic analysis of plasma conducted as part of our study revealed a multitude of alterations in protein expression patterns following CPET in patients with HF. An enrichment analysis identified some pathways that are typical for exercise-induced microinjury (such as regulation of the response to wounding and healing), which are also observed in healthy individuals [3]. However, more importantly, it linked the observed changes with processes that could potentially impact the clinical outcomes of HF such as coagulation disorder and JAK/STAT3 pathway. A change in the regulation of proteins or pathways associated with poor prognosis in patients with HF in response to an acute bout of exercise (CPET in the study) may be indicative not only of advancement of the HF, but also of overexertion. The proteins that are significantly either up- or downregulated during intensive exercise should be subjected to further investigation as potential markers for monitoring the adjustment of training intensity.

The objective of CPET is for the patient to perform a submaximal exercise. A high exercise load can activate physiological mechanisms that are not expressed under resting conditions. Consequently, the expression of proteins that are predictors of patient prognosis is anticipated to undergo alterations in response to stress conditions. The discovery that the PI3K-Akt pathway plays a role in the response to intense exercise in HF patients provides an illustrative example of the second point. Furthermore, the PI3K-Akt pathway exemplifies the advancement of knowledge regarding biomarkers and pathophysiological mechanisms, ultimately facilitating the development of novel treatment options. Recently, Salubris Biotherapeutics presented the results of a phase 1b clinical trial of JK07—first investigational antibody fusion protein and first selective ErbB4 agonist which consists of an active polypeptide fragment of the human growth factor neuregulin [NRG-1] [20]. In an experimental model, NRG-1 was shown to protect the heart against ischemia/reperfusion injury through a PI3K/Akt-dependent mechanism [21]. The clinical trial [20] revealed a significant improvement of LVEF in patients with HF with reduced LVEF six months following the administration of a single dose of JK07. Further investigation of the drug is planned as a phase 2, randomized, double-blind, placebo-controlled, multiple dose study (NCT06369298).

### 5.2. Limitations of the Study

The study group, although very homogeneous, is relatively small. To confirm the results of the study, further research should be performed on a large population targeting the predefined proteins.

The study samples were collected before the common implementation of flosins and sacubitril/walsartan into the therapy of patients with HF. As there are two groups of drugs that have spectacularly improved the prognosis of patients, their impact on the acute response to exercise in HF patients cannot be excluded.

## 6. Conclusions

Changes in the proteomic profile revealed a complexity of exercise-induced processes in patients with HF, suggesting that few major physiological pathways are involved. Further studies focusing on specific pathways are needed.

## Figures and Tables

**Figure 1 biomedicines-12-02267-f001:**
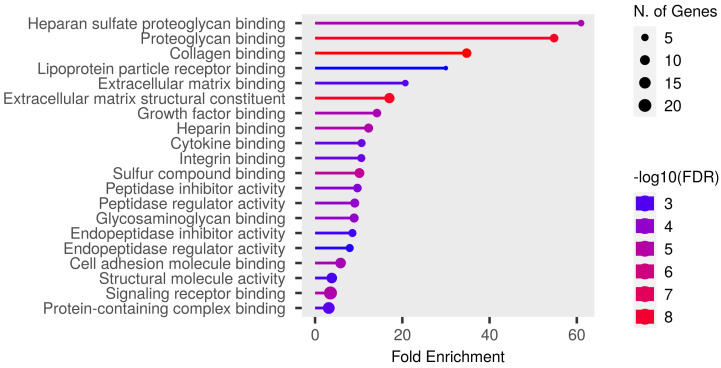
Enrichment analysis: molecular function.

**Figure 2 biomedicines-12-02267-f002:**
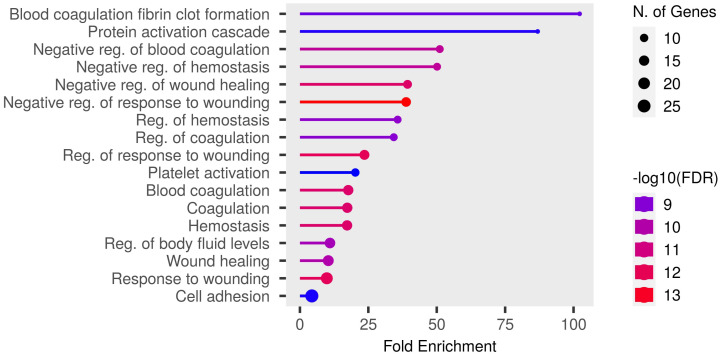
Enrichment analysis: biological processes.

**Figure 3 biomedicines-12-02267-f003:**
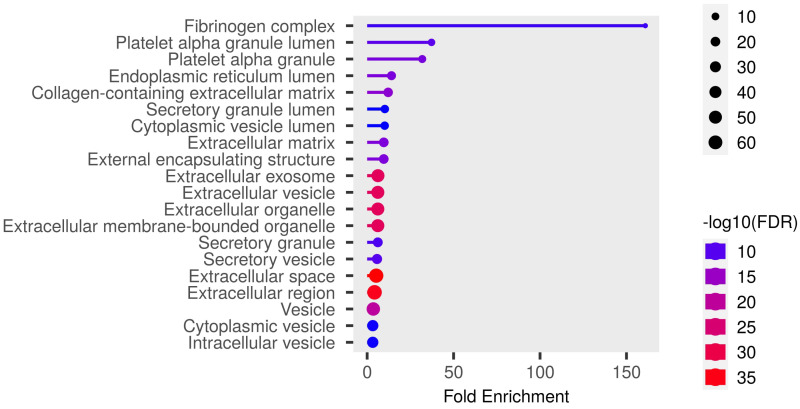
Enrichment analysis: cellular component.

**Figure 4 biomedicines-12-02267-f004:**
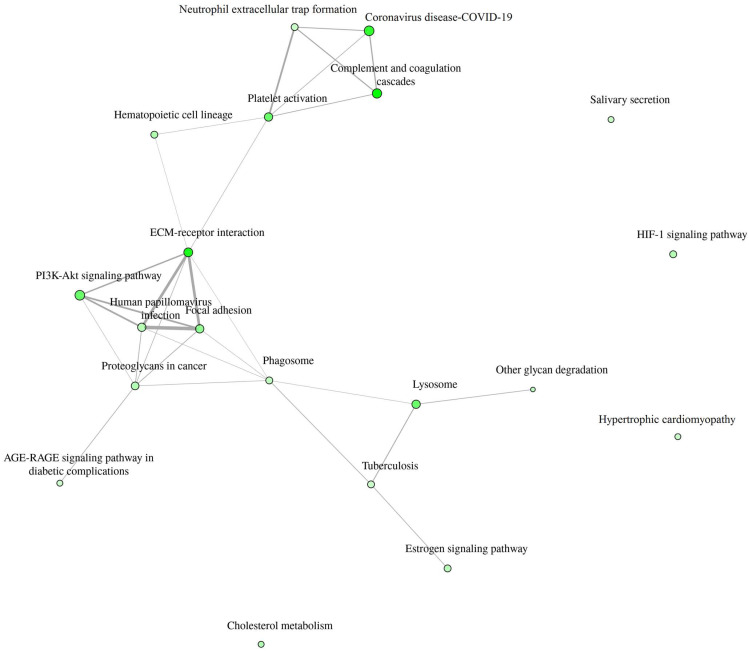
Interactive enrichment network.

**Table 1 biomedicines-12-02267-t001:** Clinical characteristics of the study population. ACEI = angiotensin-converting enzyme inhibitor, ARB = angiotensin receptor blocker, ARNI = angiotensin receptor-neprylisin inhibitor, COPD = chronic obstructive pulmonary disease, CRT-D = cardiac resynchronization therapy defibrillator, eGFR = estimated glomerular filtration rate, HF = heart failure, ICD = implantable cardioverter defibrillator, LVEDD = left ventricular end-diastolic dimension, LVEF = left ventriclar ejection fraction, NYHA = New York Heart Association, MRA = mineralocorticoid receptor antagonist, NT-pro-BNP = N-terminal prohormone of brain natriuretic peptide, pVO2 = peak oxygen uptake, RER = respiratory exchange ratio, TAPSE = tricuspid annular plane systolic excursion, VE/VCO_2_ = minute ventilation/carbon dioxide production * mitral or tricuspidal regurgitation at least moderate.

	Study Population (n = 20)
Clinical characteristics	
Age, years	53.3 ± 8.3
Gender, % male	20 (100)
Ischemic etiology of HF	10 (50)
Atrial fibrillation	11 (55)
Hypertension	7 (35)
Diabetes mellitus	9 (45)
COPD	2 (10)
ICD/CRT-D	12 (60)/6 (30)
II NYHA class, patients (%)	1 (5)
II/III NYHA class, patients (%)	4 (20)
III NYHA class, patients (%)	15 (75)
Echocardiographic parameters	
LVEF [%]	22.2 ± 6.2 (range:12.5–35%)
LVEDD [mm]	73.5 ± 9.6
TAPSE [mm]	18.4 ± 2.4
Mitral regurgitation *, patients (%)	15 (75)
Tricuspid regurgitation *, patients (%)	10 (50)
Laboratory test results	
NT-pro-BNP [pg/mL], median (interquartile range)	3187 (2011–4887)
eGFR, mL/min/1.73 m^2^	60.1 ± 14.1
bilirubin [mmol/L]	22.4 ± 11.7
Medication	
Beta-blockers [n, %]	20 (100.0)
ACEI/ARB/ARNI [n, %]	20 (100.0)
Spironolactone/eplerenone [n, %]	20 (100)
Loop diuretics [n, %]	20 (100)
More than one diuretic administered (except from MRA) [n, %]	10 (50)
Cardiopulmonary exercise test results	
pVO2 [mL/kg/min]	10.5 ± 3.3
pVO2 adjusted for sex and age adjusted pVO2 [%]	34.5 ± 10.9
RER at peak exhaustion	1.1 ± 0.1
VE/VCO_2_ slope	45.2 ± 11.3

## Data Availability

Data are available in Appendix A and additional data are available upon request.

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
