# Peer review of "Exercise-Induced Proteomic Profile Changes in Patients with Advanced Heart Failure"

_biomedicines, 2024, doi:10.3390/biomedicines12102267_

Round 1

Reviewer 1 Report

Comments and Suggestions for Authors

Overall analysis

The aim of the article is to analyse the changes in the plasma proteomic profile of HF patients after 15 cardiopulmonary exercise test (CPET) to define pathways involved in response to exercise. The main strenght of the article is the novelty and the interesting topic. The main limitation is the needed to indicate in a better modality the plasma proteins evaluate.

Section by section analysis

Title

No suggestions

Abstract

I suggest to indicate in Methods the mean age of the population, type of study and information about statistical analysis.

Introduction

No suggestions

Material and methods

In a prospective study the description of population should be insert in the results. Moreover the reference with a previous study is not a good idea. I suggest to report the inclusion and exclusion criteria. Morever the prospective study obtain the approvation of local IRB? Why did you decide to collect blood sample after 10-15 minutes?

Results

The proteins evaluted shuold be describe in material and methods. 

Discussion

I suggest to improve the section with the possible hypothesis about the change in proteins expressions. Are their related with the consequences of HF?

Author Response

We are grateful to the reviewers for dedicating their time and providing insightful feedback.

The responses to the reviewer comments are presented herewith.

  1. The aim of the article is to analyse the changes in the plasma proteomic profile of HF patients after 15 cardiopulmonary exercise test (CPET) to define pathways involved in response to exercise. The main strenght of the article is the novelty and the interesting topic. The main limitation is the needed to indicate in a better modality the plasma proteins evaluate.

Response: In order to provide a more comprehensive analysis, we have expanded the discussion to include a commentary on another protein, as well as a dedicated paragraph “Clinical implications”

2. Abstract

I suggest to indicate in Methods the mean age of the population, type of study and information about statistical analysis.

Response: We changed the “methods” section of the abstract:

Methods: The study prospectively enrolled 20 male patients with advanced HF (aged 53.3±8.3 years). Blood samples were taken from the patients before and immediately after the CPET to obtain plasma proteomic profiles. Two-sample t-tests (paired or non-paired) were performed with and without false discovery rate (FDR) correction for multiple testing. Enrichment analysis was performed to associate biological processes and pathways with the study results

3. Material and methods

In a prospective study the description of population should be insert in the results.

Response: We changed the Methods and Results section based on reviewer suggestions. Moreover the reference with a previous study is not a good idea. I suggest to report the inclusion and exclusion criteria.

The exclusion criteria were already listed:

“Therefore, the exclusion criteria were as previously described [9]: administration of catecholamines, contraindications to performance of cardiopulmonary exercise test (CPET), pneumonia or bronchitis within last two weeks, or severe ventilation disorders with forced expiratory volume in one second (FEV1) < 50%.”

We additionally enlisted inclusion criteria:

“We randomly selected 10 men with ischemic and 10 men with non-ischemic etiologies of HF from a population of 51 patients examined in our earlier study [9], the inclusion criteria of which were: patients aged 18-70 years with advanced heart failure with reduced left ventricular ejection fraction (LVEF; less than 40%) who have clinical indication to undergo cardiopulmonary exercise test (CPET) and right heart catheterization.

4. Morever the prospective study obtain the approvation of local IRB?

Response: The information was given (according journal guidelines and template in the end of the paper). Additionally we have added the number of approval

“Institutional Review Board Statement

The study was conducted in accordance with the Declaration of Helsinki and approved by the Regional Ethics Committee of National Institute of Cardiology (agreement number 1928).”

5. Why did you decide to collect blood sample after 10-15 minutes?

Response: We discussed the issue in the first paragraph of the Discussion.

“In designing our study, we had to address some important methodological issues that could significantly affect the results. The first was to determine the optimal time to collect a blood sample after exercise. As shown by Mi et. al [6], the change in the proteome induced by exercise decreases with time, and one hour after peak exercise only 7% of the proteins still present a changed expression. At the same time, new sets of proteins with altered expression are discovered, but they are not as numerous. Because we wanted to define the response to acute exercise in HF patients, we decided to take blood samples during the resting period, no later than 15 minutes after completion of the CPT. In this way, we minimized the risk of unanticipated injury associated with needle punctures (this can happen when taking blood while exercising or while standing) and were able to detect the majority of proteomic changes.”

6. Results

The proteins evaluted shuold be describe in material and methods. 

Response: We’ve moved to the Methods section:

3.2 Protein identification

A total of 968 plasma proteins were identified, of which 772 were suitable for statistical analysis. The complete list of the 772 proteins that were subjected to analysis can be found in the Supplementary File 1.”

Moreover, we have added to the Excel file a sheet with the list of all analyzed proteins.

7. Discussion

I suggest to improve the section with the possible hypothesis about the change in proteins expressions. Are their related with the consequences of HF?

Response: In order to provide a more comprehensive analysis, we have expanded the discussion to include a commentary on another protein, as well as a dedicated paragraph “Clinical implications”

Reviewer 2 Report

Comments and Suggestions for Authors

Overall, this manuscript makes valuable contributions to its field. Proteomic profile changes in heart failure patients after exercise suggest involved physiological pathways. Some of these findings are consistent with previous studies, but further studies with larger samples of patients are needed. Certain aspects of its physiological interpretation of involved pathways could be strengthened to enhance its scientific impact and readability.

Author Response

We are grateful to the reviewers for dedicating their time and providing insightful feedback.

In order to provide a more comprehensive analysis, we have expanded the discussion to include a commentary on another protein, as well as a dedicated paragraph “Clinical implications”